# Timing urinary tract reconstruction in rats to avoid hydronephrosis and fibrosis in the transplanted fetal metanephros as assessed using imaging

**Kotaro Nishi**[1], **Takafumi Haji**[1], **Takuya Matsumoto**[1], **Chisato Hayakawa**[1],
**Kenichi Maeda**[1], **Shozo Okano**[1], **Takashi Yokoo**[2,3], **Satomi Iwai**[1,3]*

**1** Laboratory of Small Animal Surgery 2, School of Veterinary Medicine, Kitasato University, Towada, Aomori, Japan, **2** Division of Nephrology and Hypertension, Department of Internal Medicine, The Jikei University School of Medicine, Minato-ku, Tokyo, Japan, **3** Meiji University International Institute for Bio-Resource Research, Kawasaki, Kanagawa, Japan

* iwai@vmas.kitasato-u.ac.jp

**Data Availability Statement:** All relevant data are within the manuscript and its Supporting information files.

## Abstract

Chronic kidney disease leads to high morbidity rates among humans. Kidney transplantation is often necessary for severe symptoms; however, options for new curative treatments are desired because of donor shortage. For example, it has been established that the kidneys can efficiently generate urine after transplantation of the metanephros, ureter, and bladder as a group. After transplantation, the urine can indirectly flow into the recipient's bladder using a stepwise peristaltic ureter system method where the anastomosis is created via the recipient's ureter for urinary tract reconstruction. However, the growth of the regenerated metanephros varies significantly, whereas the time window for successful completion of the stepwise peristaltic ureter system that does not cause hydronephrosis of the metanephros with bladder (ureter) is quite narrow. Therefore, this study was conducted to periodically and noninvasively evaluate the growth of the transplanted metanephros, ureter, and bladder in rats through computed tomography and ultrasonography. The ultrasonographic findings highly correlated to the computed tomography findings and clearly showed the metanephros and bladder. We found that the degree of growth of the metanephros and the bladder after transplantation differed in each case. Most of the rats were ready for urinary tract reconstruction within 21 days after transplantation. Optimizing the urinary tract reconstruction using ultrasonography allowed for interventions to reduce long-term tubular dilation of the metanephros due to inhibited overdilation of the fetal bladder, thereby decreasing the fibrosis caused possibly by transforming growth factor-β1. These results may be significantly related to the long-term maturation of the fetal metanephros and can provide new insights into the physiology of transplant regeneration of the metanephros in higher animals. Thus, this study contributes to the evidence base for the possibility of kidney regeneration in human clinical trials.

**Funding:** The author(s) received no specific funding for this work.

**Competing interests:** The authors have declared that no competing interests exist.

## Introduction

The morbidity rate of end-stage kidney disease remains high. Although kidney transplantation is the main curative treatment, securing an appropriate donor is difficult [1]. In addition to the number of people waiting for a kidney transplant, the number of patients undergoing hemodialysis is increasing. Since there are many individuals with end-stage kidney disease worldwide, the associated treatment often drastically increases household medical expenses, imposing a heavy burden on families in particular and society in general [2].

There is a need for alternative treatment options for kidney transplantation and dialysis, as the former is invasive, and the latter is unsustainable in the long term. Ideally, the alternative must be fundamentally curative. For example, there has been an attempt to develop a kidney using pluripotent stem cells. Some studies report experimentation with a method to produce kidney or nephron and ureteric bud progenitor cells *in vitro* by using human stem cells [3, 4]. Organoids that have successfully matured within *in vitro* cultures include the differentiating nephrons, interstitium, and vasculature. In addition, the use of human stem cells has made it possible to produce organoids that resemble human fetal kidneys. However, the resulting kidneys were too small to be utilized and could not produce urine due to the immaturity of the tubules, glomerular neovascularization, and urinary excretion pathway [3].

To overcome these issues, we attempted to generate a kidney from mesenchymal stem cells using a nephrogenic niche in xeno-animals as the scaffold [5–7]. By eliminating the native nephron progenitor cells (NPCs) from the nephrogenic zone during development, nephrons from external NPCs can be successfully generated [8, 9]. It has previously been confirmed that this system can generate interspecies chimeric nephrons between rats and mice [9]. Furthermore, the use of induced pluripotent stem cells, as well as mesenchymal stem cells, has been considered for this purpose. NPCs differentiated from induced pluripotent stem cells obtained from hemodialysis patients can be used without cell quality deterioration when compared to those from healthy controls; moreover, differentiation into glomeruli with the new blood vessels has been confirmed by transplanting to kidneys in mice [10].

Based on these successes, this study aimed to scale-up the experiment, using bigger animals to determine the efficacy and safety of these organ components for human clinical use. However, transplantation alone does not provide a route for excretion of the produced urine. Thus, the metanephros-only transplant can cause hydronephrosis and kidney insufficiency [11–13]. This may be solved using a stepwise peristaltic ureter (SWPU) system (S1 Fig), which comprises the anastomosis of the recipient ureter to a developed metanephros with bladder (MNB) in rats [11]. This new urinary tract reconstruction method allows continuous excretion of urine produced from the MNB into the recipient's bladder via the recipient ureter [11]. However, the optimal timing for the SWPU after transplantation is crucial and depends on differences in the growth of the MNB to avoid hydronephrosis [11, 12]. Postrenal nephropathy due to hydronephrosis and the delayed relief from the obstruction have substantial effects on the kidneys [14–16]. Furthermore, the ureteral primordial obstruction during the fetal stage has been shown to cause dysplastic metanephros [17]. Therefore, we believe that early relief of any obstruction significantly improves subsequent kidney functions, even with fetal-derived grafts.

In cases of xenograft or MNB transplantation in large experimental animals such as pigs, dogs, or humans, the effects of individual differences are considered to be significant and may indirectly determine the outcome. Therefore, the appropriate time for SWPU should be determined using a minimally invasive method that can be used to frequently observe the MNB progress in a clinical setting. In general practice, imaging methods, including contrast-enhanced computed tomography (CT) and particularly ultrasonography, have been used to easily assess the body condition while being minimally invasive.

Therefore, in this study, we aimed to establish an appropriate time index for SWPU intervention using morphological and histopathological examination through image analysis in the clinical setting.

## Materials and methods

### Animals

The animal rearing management was in accordance with the Kitasato University Faculty of Veterinary Medicine Animal Experiment Guideline and Manual for Rearing and Management of Experimental Animals. This experiment was approved by the Animal Care and Use Committee of Kitasato University (Approval No: 17–127, 18–127, 19–085). All surgery was performed under isoflurane anesthesia, and all efforts were made to minimize suffering. The rats were housed in cages under temperature- and light-controlled conditions in a 12:12-hour light: dark cycle and were provided with fresh food and water *ad libitum*.

In Phase 1, we used three pregnant female Lewis rats on embryonic Day 15 (E15) (Japan Charles River Laboratories, Kanagawa, Japan) to obtain the fetal MNB. As recipient rats (organ recipient animals), we used 12 male Lewis rats (Japan Charles River Laboratories) aged 11 weeks, with a mean bodyweight of 309.0 ± 11.4 g.

In Phase 2, we used three pregnant female Lewis rats on E15 (Japan Charles River Laboratories) to obtain the fetal MNB. Eighteen male Lewis rats (Japan Charles River Laboratories) aged 9 weeks with a mean bodyweight of 243.0 ± 7.5 g were used as recipient rats.

In Phase 3, we used three pregnant female Lewis rats on E15 (Japan Charles River Laboratories) to obtain the fetal MNB. Nine male Lewis rats (Japan Charles River Laboratories) aged 10 weeks with a mean bodyweight of 292.8 ± 7.6 g were used as recipient rats.

### Isolation and grafting of the MNB

The surgery was performed by an experienced surgeon specialized in microsurgery. Pregnant rats were anesthetized with 2.5% isoflurane inhalation. The embryos were harvested, and the pregnant rats were then sacrificed immediately via an infusion of pentobarbital (120 mg/kg). All the embryos were euthanized via decapitation. The MNBs were dissected under a surgical microscope, as previously described [11].

### Phase 1: Imaging evaluation

The flow of the Phase is shown in Fig 1A. Anesthesia was introduced and maintained in the recipient rats using 2.5% isoflurane inhalation. After performing a laparotomy through a midline abdominal incision, the intestinal tract was pulled out of the body, and the retroperitoneum and the abdominal aorta were exposed. Observing the animal under a surgical microscope, a small incision was made to the retroperitoneum, and a first MNB (MNB1) was transplanted in the retroperitoneal space near the abdominal aorta. The day of the transplant operation was considered Day 0. After transplantation, a single interrupted suture was made on the retroperitoneum using a 6–0 non-absorbable suture thread (PROLENE®, Johnson and Johnson K.K., Tokyo, Japan). The wound was closed using conventional methods. The animals with the transplanted MNB were divided into two groups. The first group comprised randomly selected rats that had the left recipient kidney removed on Day 28 after MNB1 transplantation and those that underwent SWPU by anastomosing the recipient ureter to the MNB1 (SWPU group; n = 8). There was urine retention in the MNB bladder, and the bladder volume of MNB when SWPU was performed was in the range of 0.016 to 0.178 cm$^3$. The second group consisted of randomly selected rats that did not undergo SWPU (non-SWPU

**A    Phase 1**

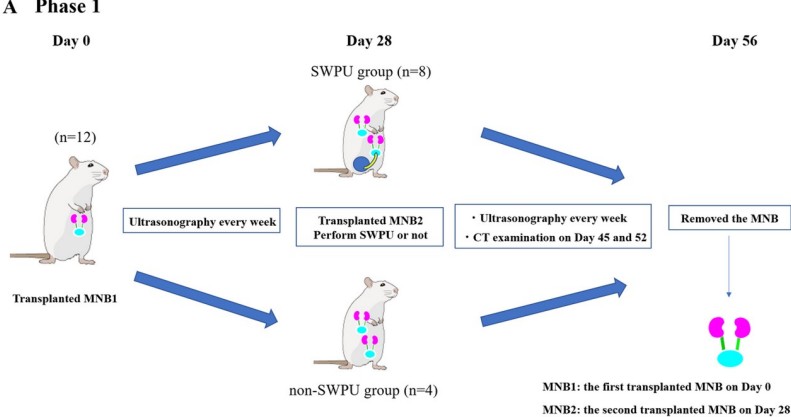

**B    Phase 2**

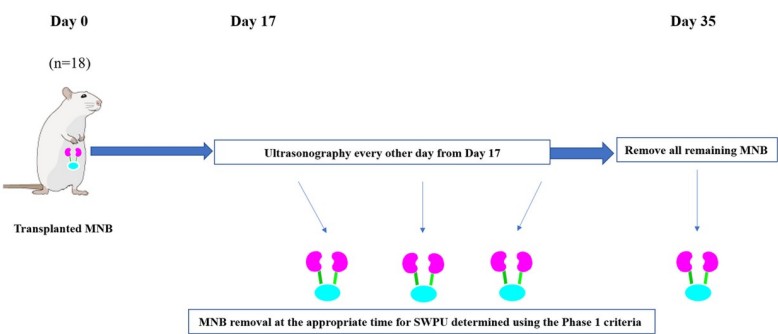

**C    Phase 3**

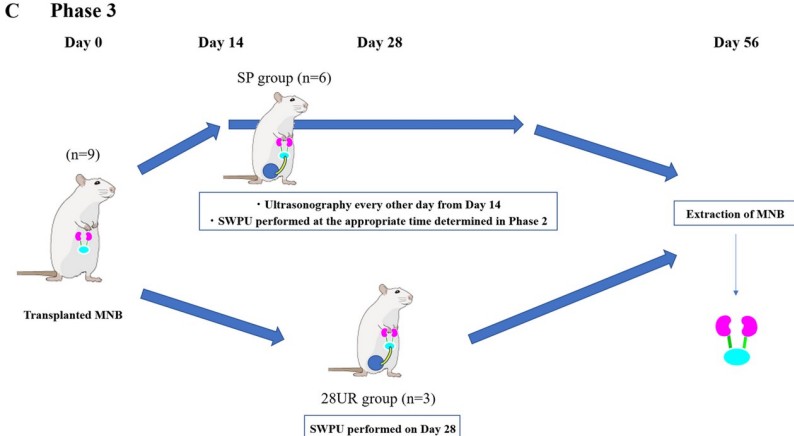

**Fig 1. A.** The Phase 1 flow chart. The MNBs were transplanted to the recipient rats on Day 0 and the MNBs were evaluated each week from Day 7 until Day 56 (Week 8) using ultrasonography. Contrast CT was performed on Days 45 and 52. **B.** The Phase 2 flow chart. Only one MNB was transplanted and evaluation of the MNB was performed using ultrasonography from Day 17. **C.** The Phase 3 flow chart. Only one MNB was transplanted into a recipient on Day 0. The MNB was evaluated using ultrasonography from Day 14. SWPU was performed if the subject met the criteria determined in Phase 2 (SP group). In the other group, the rats underwent SWPU for urinary tract reconstruction on Day 28 (28 UR). MNBs in both groups were removed on Day 56.

group; n = 4). Both groups underwent a second MNB (MNB2) transplant on Day 28. The MNB2 transplant was distinguished from MNB1 by locating the MNB2 to the head side of MNB1. Ultrasound examination was performed every week starting from Day 0 to Day 56 (Week 8). Furthermore, ultrasound and contrast-enhanced CT examinations were performed on Days 45 and 52. On Day 56 after MNB1 transplantation, both MNB1 and MNB2 were removed, and histopathological examinations were conducted.

## MNB assessment using ultrasonography

Three experienced technicians who practice ultrasonography daily examined randomly selected rats. To carry out the examinations, the rats were anesthetized, and the anesthesia was maintained until the end of the procedure using 2.5% isoflurane inhalation. The animals' abdomens were shaved, and they were laid in the supine position. A LOGIQ S8 ultrasound machine (GE Healthcare Japan K.K., Tokyo, Japan) was used. After identifying the MNB through the observation of axial, sagittal, and coronal cross-sections, maximum long-axis length of the sagittal cross-section (L), maximum short-axis width of the axial cross-section (W), and height of the maximum depth (H) were measured. The volume ($V$) of the MNB bladder was also assessed.

As a probe, we used a 3–11 MHz linear array. We used the Color Doppler imaging mode to identify the presence/absence of blood flow to the transplanted MNB. To measure the MNB size and volume and to observe its morphology, we used B mode and set the gain and depth to 90 and 2.3–2.5 cm, respectively. Volume calculation using ultrasonography assumed the MNB to be a spheroid, substituting the measured values into the formula shown below for the volume of a spheroid:

$$V = \frac{\pi}{6} \times \text{L} \times \text{W} \times \text{H}$$

## MNB assessment using contrast CT

For imaging, we used the Aquilion 16 Multi-slice CT system (Toshiba Medical Systems K.K., Tochigi, Japan). The imaging was conducted in dynamic CT mode, at a tube voltage of 80 kV, tube current of 150 mA, an imaging rotation speed of 0.5 sec/rotation, and a slice thickness of 0.5 mm. The radiation exposure dose was kept unified at 392.7 mGy. The animals were anesthetized, with the anesthesia maintained with 2.5% isoflurane inhalation on Days 45 and 52 (2.5 weeks and 3.5 weeks after MNB2 transplantation, respectively), and kept in the supine position during imaging.

For contrast CT examinations, each animal received a bolus injection of 0.3 mL iohexol (Omnipaque® 300 Injection, Daiichi Sankyo K.K., Tokyo, Japan) into the tail vein. Images were taken 30 min after administering the contrast agent to allow its accumulation in the MNB.

After reconstitution and reconstruction of the images, they were processed using OsiriX DICOM Viewer software (Pixmeo SARL, Geneva, Switzerland) to assess the morphological characteristics and MNB volume.

## Phase 2: Timing of SWPU

The flow of this Phase is shown in Fig 1B. Only a single MNB transplantation was performed using the same methods as described for Phase 1 (Day 0). From Phase 1, we established that the appropriate time for urinary tract reconstruction would be when a single vacuole is

observed in the MNB and when the bladder volume of MNB becomes 0.016 cm$^3$, which is the minimum practicable volume of SWPU.

For the SWPU, an experienced operator who routinely used microsurgical techniques conducted the urinary tract reconstruction according to the methods described previously [11]. A previous report showed that the tubules of the MNBs were less dilated when SWPU was performed than when SWPU was not performed; the patency between the MNB bladder and the recipient ureter was confirmed with contrast-enhanced CT scan and the continuity of the transitional epithelium at the anastomotic site was confirmed using histopathological examination [11]. Therefore, in Phase 2, the MNB was removed based on the size of the smallest MNB bladder volume possible for SWPU, confirmed using the ultrasound and histopathological examinations obtained in Phase 1. The MNB observations were performed every other day starting from Day 17 after transplantation. The morphological characteristics and the MNB volume were assessed using the same technique as in Phase 1. If the MNB met the criteria for SWPU, as established in Phase 1, the animals were euthanized and the MNB was removed. Animals that did not achieve the criteria were observed until Day 35 before having the MNB removed. The removed MNB was fixed for histopathological examination.

## Phase 3: Maturity of the MNB associated with the timing of SWPU

The flow of this Phase is shown in Fig 1C. Only a single MNB transplantation was performed using the same methods as described for Phase 2. The MNB observations were performed every other day from Day 14. The morphological characteristics and MNB volume were assessed with the same technique used in Phase 1. Six animals were randomly selected and if the MNBs met the criteria for appropriate timing of urinary tract reconstruction, as established in Phase 2, they underwent SWPU (SP group; n = 6). The remaining three randomly selected animals underwent SWPU for urinary tract reconstruction on Day 28 (28UR group; n = 3) and were observed until Day 56 (Week 8). After measuring the glomerular filtration rate (GFR), the MNBs of rats were removed and fixed for histopathological examination. The rats were euthanized after the removal of the MNBs.

## Removal of the transplanted MNB

After anesthetizing the rats with 2.5% isoflurane inhalation, we made a midline abdominal incision and removed the MNBs, which were then used for histopathological examination. After removal, we intraperitoneally administered 125 mg/kg of pentobarbital and confirmed cardiopulmonary arrest 15 min later.

## Histopathological examination

The MNB tissue was fixed using 4% paraformaldehyde in phosphate buffer solution and embedded in paraffin as previously described [11]. Hematoxylin-eosin (HE) dye and Masson's trichrome (MT) staining were used in all Phases. In Phase 3, the tissues were thinly sliced to 2 μm thickness and stained for transforming growth factor-β1 (TGF-β1: sc-130348, Santa Cruz Biotechnology, Santa Cruz, CA), collagen-α1 type 1 (sc-293182, Santa Cruz Biotechnology, Santa Cruz, CA), and vimentin (422101, Nichirei Biosciences Inc., Tokyo, Japan). Terminal deoxynucleotidyl transferase dUTP nick end labeling (TUNEL) assay was performed using the In-Situ Apoptosis Detection Kit (Takara Bio Inc., Shiga, Japan) according to the manufacturer's instructions. The entire metanephros cut at maximum length was observed at 400× magnification. One slide was randomly selected to acquire images and all tests were blindly observed from multiple slides by three people. HE, MT, TGF-β1, collagen-α1 type 1, and vimentin measures were based on 20 non-overlapping images taken and calculated based on the average

area. For evaluation using TUNEL staining, apoptosis of the tubular and glomerular cells was counted in 10 non-overlapping images observed at 400× magnification and the ratio of these cells to unstained cells was calculated. The images were assessed by researchers who were blinded to the study; ImageJ$^{®}$ analysis software (National Institutes of Health, Bethesda, Maryland, USA) was used for assessment. HE-stained slides were used to evaluate the tubular lumen area and MT-stained slides were used to evaluate the interstitial fibrosis in the metanephros.

## GFR measurement

The GFR was measured using a commercially available kit (Diacolor$^{®}$ Inulin, Toyobo Co., Osaka, Japan) by following the kit instructions. On Day 56 after the transplantation in Phase 3, bilateral nephrectomy was performed under the same anesthesia as mentioned previously. The blood was collected from the tail vein of the rats at 1 and 2 hours after inulin administration, and the GFR measurement was obtained using the plasma. Normal kidney GFR values were obtained from healthy adult rats (S1 Table).

## Statistical analyses

The results are presented as mean ± standard deviation. All statistical analyses were performed with EZR (Saitama Medical Center, Jichi Medical University, Saitama, Japan) [18], a modified version of R commander designed to add statistical functions frequently used in biostatistics. Scatter plots and Pearson's product ratio correlation coefficients were used to compare calculated urinary volume using ultrasonography and contrast CT findings and the storage of the urine actually collected. Scatter plots and Pearson's product ratio were used to determine the relationship between TGF-β1 expression levels and the percentage of apoptotic cells as well. The paired Student's t-test was used to examine the difference in the ultrasonographically observed volumes of MNB observed over time. The Mann–Whitney U test was used to analyze tubular dilation and metanephros fibrosis was examined through histopathological examination. A $p$-value of 0.05 was considered statistically significant.

## Results

### Phase 1: Imaging evaluation

Phase 1 was a morphological assessment of two transplanted MNBs using contrast CT and ultrasonography. The first transplanted MNB was designated as MNB1 (n = 12) and the second transplanted MNB was designated as MNB2 (n = 12). Furthermore, the group in which SWPU was carried out in MNB1 on Day 28 after transplantation was set as the SWPU group (n = 8), and the group in which it was not carried out was set as the non-SWPU group (n = 4).

**MNB detection rate using contrast CT and ultrasonography.** Contrast CT and ultrasonography allowed for the evaluation of all MNBs by Day 21 after transplantation (Fig 2). However, it was not possible to assess the MNB or bladders with hydronephrosis. The ultrasonographic examination allowed for early-stage detection of the MNB that was present near the aorta, with clear margins. The ultrasonographic examination also allowed for 100% recognition of both the MNB1 (the first transplanted MNB on Day 0) and MNB2 (the second transplanted MNB on Day 28) (Table 1). The measured volume of MNB is shown in Table 2. The MNB that seemed to be normal was confirmed to have only a single vacuole, and the MNB that had hydronephrosis was observed to have multiple vacuoles. (Fig 2B and 2C). Additionally, the partial observation of the newly formed blood vessels around the MNB was made using the Color Doppler method (Fig 2D).

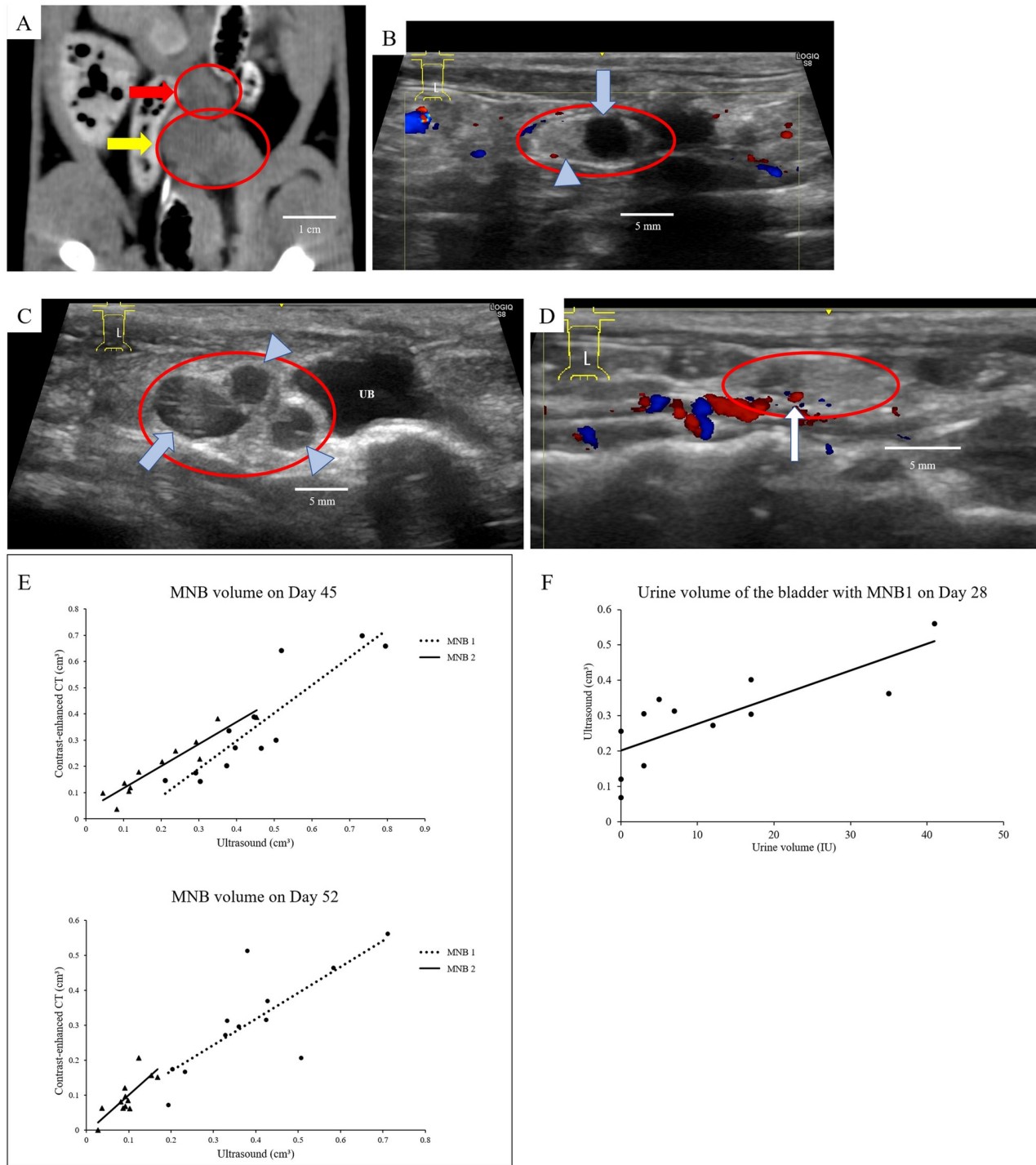

**Fig 2. A.** Imaging using computed tomography. Visualization of the MNBs on the abdominal arteriovenous vein under the retroperitoneal area using contrast-enhanced CT. (red arrow: MNB1, yellow arrow: MNB2, red circle: MNB). **B.** Imaging of the MNB when it was considered normal using ultrasonography. Ultrasonographic images of urine retention in the MNB bladder indicating normally maturing MNB (blue arrow: bladder of MNB, arrowhead: metanephros of MNB). Only one vacuole was found in MNB. **C.** Imaging of the MNB using ultrasonography when hydronephrosis was suspected. Ultrasonography image of suspected hydronephrosis of the MNB (UB: Recipient's bladder). Three vacuoles in MNB were identified and were of approximately the same size as the recipient's bladder. **D.** Imaging of blood flow around the MNB. The blood flow around the MNB was confirmed using Color Doppler in ultrasound (white arrow: blood flow in MNB). **E.** Correlation of MNB volumes using ultrasonography and CT. Correlation between the MNB volumes detected using weekly ultrasonography and contrast CT inspection during Phase 1. The dotted line represents

the MNB1 and the solid line represents the MNB2. Upper figure: MNB volume on Day 45 (MNB1, R = 0.78; MNB2, R = 0.79). Lower figure: MNB volume on Day 52 (MNB1, R = 0.90; MNB2, R = 0.94). **F.** Correlation between urine volume measured using ultrasound and urine volume actually collected. Correlation between urine volume accumulated in MNB1 bladder and actual urine volume on Day 28 estimated using ultrasound findings (R = 0.79).

**Table 1. MNB detection rate using ultrasonography from post transplantation in Phase 1.**

|  | Detection rate of MNB after transplantation | | | |
|---|---|---|---|---|
|  | **Day 7** | **Day 14** | **Day 21** | **Day 28** |
| **MNB1 (n = 12)** | 16.7% | 83.3% | 100% | 100% |
| **MNB2 (n = 12)** | 75% | 91.7% | 100% | 100% |

MNB: metanephros with bladder (ureter); MNB1: the first transplanted MNBs; MNB2: the second transplanted MNBs.

The ultrasound finding-based detection rates of MNB1 and MNB2 after transplantation are shown in the Table. Day: days after transplantation of each MNB.

**Correlation between retained urine volume in MNB1 and MNB bladder volume.** The comparison of MNB volumes measured using contrast CT and ultrasonography indicated a strong positive correlation (Day 45: MNB1, R = 0.78; MNB2, R = 0.79 and Day 52: MNB1, R = 0.90; MNB2 = 0.94) (Fig 2E). The amount of urine collected in the bladder of MNB1 on Day 28 showed a strong positive correlation with the MNB volume determined using ultrasonography (R = 0.79) (Fig 2F).

**Tubular dilation and fibrosis in MNB1 and MNB2 assessed using a histopathological examination.** The MNB1 tubular dilation was significantly larger in the non-SWPU group than in the SWPU group ($p < 0.05$) (data not shown). For the MNB2 in both groups, there was tubular dilation; nevertheless, no significant difference was observed in the tubular dilation regardless of the SWPU in the MNB1. Additionally, there was no significant difference in fibrosis between findings of MNB1 and MNB2, irrespective of the SWPU.

**Table 2. MNB volume transition during observation using contrast-enhanced CT and ultrasonography examinations in Phase 1.**

|  |  | MNB volume in each week using imaging devices (cm³) | | | | | | | | | |
|---|---|---|---|---|---|---|---|---|---|---|---|
|  |  | **Day 7** | **Day 14** | **Day 21** | **Day 28** | **Day 35** | **Day 42** | **Day 45** | **Day 49** | **Day 52** | **Day 56** |
| CT | MNB 1 (n = 12) | — | — | — | — | — | — | 0.310 ± 0.141* | — | 0.352 ± 0.195* | — |
|  | MNB 2 (n = 12) | — | — | — | — | — | — | 0.096±0.053 | — | 0.203±0.107 | — |
| Ultrasound | MNB1 (n = 12) | ND | 0.073± 0.041 | 0.139 ± 0.090† | 0.277 ± 0.131 | 0.271 ± 0.082† | 0.396 ± 0.136 | 0.390 ± 0.148 | 0.361 ± 0.146 | 0.452 ± 0.164 | 0.381 ± 0.252 |
|  | MNB2 (n = 12) | — | — | — | — | 0.019 ± 0.010 | 0.098 ± 0.090 | 0.096 ± 0.039† | 0.145 ± 0.082† | 0.203 ± 0.120 | 0.214 ± 0.164 |

CT: computed tomography; MNB: metanephros with bladder (ureter).

The urinary volume of the MNB was measured using CT on Day 45 and Day 52, and ultrasonography was performed weekly until Day 56 (Week 8).

*$p < 0.01$ vs. Results using ultrasound were obtained on the same day,

† $p < 0.01$ vs. Previous measurement in the same MNB group (Day: days after transplantation of MNB1).

## Phase 2: Timing of SWPU

In Phase 2, we set up hypotheses to determine the appropriate time for SWPU. Phase 2 proceeded only after ultrasonographically establishing that ① hydronephrosis of the metanephros and >2 vacuoles were not observed in the MNB and ② the MNB bladder was larger than 0.016 cm$^3$. The index of this volume is the minimum volume of the MNB bladder that can undergo SWPU as calculated from Phase 1.

**Number of days from MNB transplantation to MNB excision and progression in the rate of tubular dilation and fibrosis.** The results of Phase 2 are shown in Fig 3. All MNBs were confirmed within 17 days after transplantation (Day 17). MNBs with no bladder formation or with hydronephrosis were deemed poorly developed (3/18). The mean number of days until MNB removal, not including the poorly developed ones, was 20.7 ± 3.6 days (range, 17 to 29 days) (Fig 3A). For the 72.2% (13/15) of the MNBs excised, it was deemed appropriate to perform SWPU within 21 days from transplantation. For 20% (3/15), 26.7% (4/15), 40% (6/15), and 13.3% (2/15), Days 17, 19, 21, and 28 or more, respectively, were deemed appropriate for performing SWPU. The MNBs that were excised before 21 days after transplantation had a significantly milder dilation of the tubular lumen than those removed after Day 21 ($p$ <0.01)

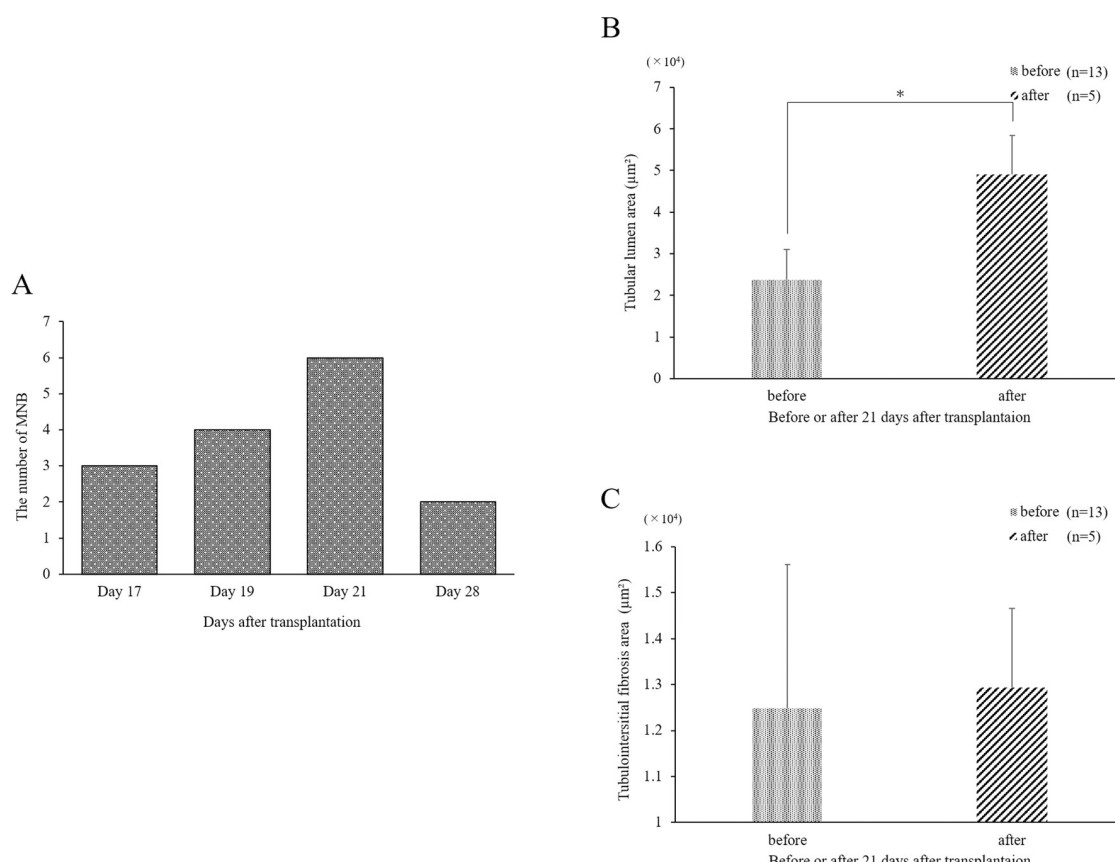

**Fig 3. Days until MNB removal and histopathological analysis.** A) The number of days until optimal SWPU after transplantation and the number of MNBs removed in Phase 2. Most of the MNBs were removed within 21 days. B) Comparison of tubular dilation in the MNBs removed before and after 21 days from transplantation. The expansion area of the tubular lumen in the removed MNBs before or on Day 21 was significantly different compared to that after 21 days from transplantation ($p$ < 0.01). C). Comparison of interstitial fibrosis in the removed MNBs before or on Day 21 from transplantation and after 21 days from transplantation.

**Table 3. GFR values in the SP and 28UR groups.**

|  | SP group (n = 6) | 28UR group (n = 3) |
|---|---|---|
| GFR (mL/min/m$^2$) | 1.95 ± 1.04 | 1.23 ± 1.22 |
| Compared to adult healthy rats (%) | 1.1 ~ 5.7% | 0 ~ 5.1% |

28UR: underwent SWPU on Day 28; GFR: glomerular filtration rate; SP: underwent stepwise peristaltic ureter (SWPU) established in Experiment Phase 2.

(Fig 3B). There was no difference between the amount of fibrosis noted in the MNBs removed before and after Day 21 (Fig 3C).

## Phase 3: Maturity of the MNB associated with the timing of SWPU

**GFR value of the MNB on Day 56 after transplantation.** Table 3 shows the GFR measurement results. The GFR was measured in all the animals in the SP (underwent SWPU established in Phase 2) and 28UR (underwent SWPU on Day 28) groups. Although no significant difference was observed between the SP and 28UR groups, none of the animals in the SP group had a GFR of 0%.

**Histopathological examination of the MNB.** An image of the extracted MNB is shown as an example (Fig 4A). In the SP group, the color of the surface of the metanephros visually confirmed blood flow, and the growth of the MNBs in the SP group occurred without hydronephrosis. As shown in Fig 4B, in the 28UR group, the observed shape of the metanephros was irregular. In some cases, the metanephros was hydronephrotic without marked liquid storage in the MNB bladder. Fig 5A shows a micrograph of HE and MT staining for evaluating tubular dilatation and interstitial fibrosis. The 28UR group tended to expand compared to the SP group; however, no significant difference was observed between the two groups (Fig 5B). A comparison of interstitial fibrosis is shown in Fig 5B. Interstitial fibrosis was significantly less in the SP group than in the 28UR group ($p$ <0.01). These results indicated that fibrosis progressed from the initial stage of tubular dilation. The measurements of the fibrosis marker are shown in Fig 5C. TGF-β1 was strongly expressed in the 28UR group, mainly in the tubular cells, interstitium, and glomeruli; similarly, vimentin and collagen-α1 type I showed significant expression in the tubular interstitium. The expression region of the fibrosis marker was

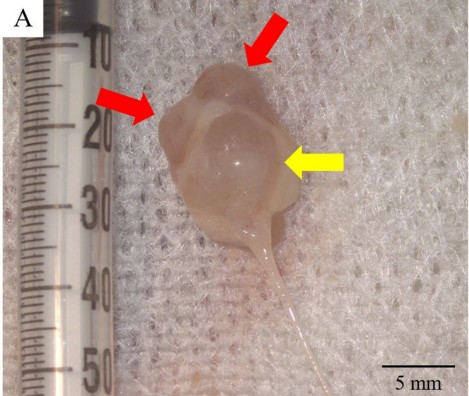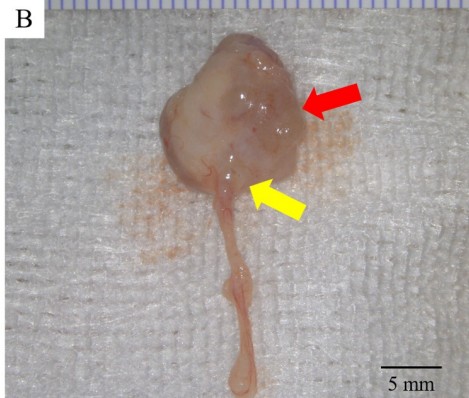

**Fig 4. Example of the extracted MNB in Phase 3.** The red arrows indicate the metanephros and the yellow arrows indicate the bladder. Fig 4A shows that the metanephros did not expand and the MNB is considered to have grown steadily. In contrast, the MNB in Fig 4B has an irregular shape with severe metanephros hydronephrosis.

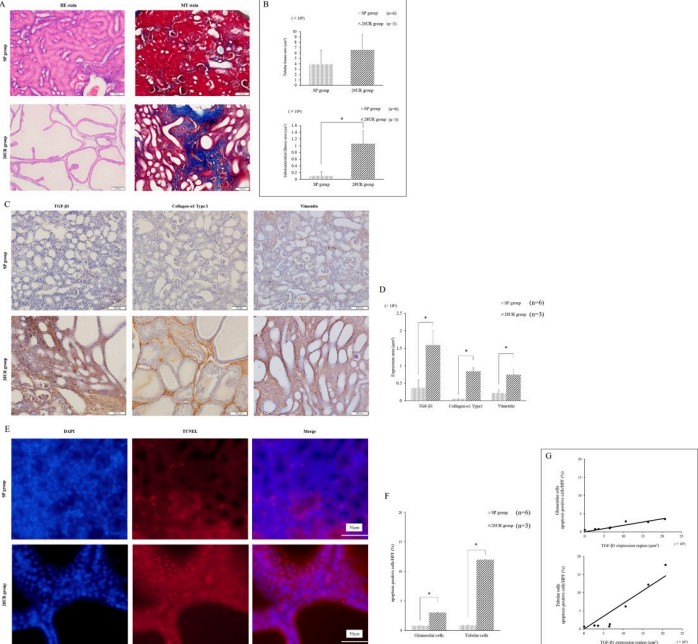

**Fig 5. Histopathological examination analysis in Phase 3.** A) The pathological tissue image using HE and MT staining of the MNB was removed 56 days after transplantation in the SP and 28UR groups. B) Comparison of the tubular lumen expansion and interstitial fibrosis areas between the SP and 28UR groups. Fibrosis is milder in the SP group than in the 28UR group. C) Expression of TGF-$\beta$, collagen-$\alpha$1 type I, and vimentin in Phase 3. Stained image of TGF-$\beta$1, collagen-$\alpha$1 type I, and vimentin in the SP and 28UR groups. D) Comparison of TGF-$\beta$1, collagen-$\alpha$1 type I, and the vimentin-presenting area between the SP and 28UR groups. The expression of these assessed factors (TGF-$\beta$1, collagen-$\alpha$1 type I, and vimentin) was significantly lower in the SP group than in the 28UR group ($p < 0.01$). E) Image of TUNEL staining in the SP and 28UR groups. F) Ratio of tubular cells and glomerular cells showing apoptosis in both groups. G) Upper figure: correlation between apoptosis rate and TGF-$\beta$1 expression region in glomerular cells; Lower figure: correlation between apoptosis rate and TGF-$\beta$1 expression region in tubular cells.

significantly lower in all evaluations in the SP group than in the 28UR group ($p < 0.01$) (Fig 5D). The image of TUNEL staining is shown in Fig 5E. The percentage of apoptotic cells was significantly lower in the MNBs of the SP group than of the 28UR group ($p < 0.05$) (Fig 5F). Furthermore, there was a strong positive correlation between TGF-β1 expression and the percentage of apoptotic cells in both the glomerular and tubular cells ($p < 0.01$) (Fig 5G).

## Discussion

Obstruction of the urinary tract during the development of rat kidneys reportedly causes developmental suppression and persistent damage after maturation [14, 15]. For this reason, it is necessary to investigate the appropriate timing for performing SWPU to avoid damage to the MNB. Therefore, in the present study, two MNBs were transplanted into suitable rat models, and their development was closely observed using ultrasonography and CT. Our results indicate that the best time for the reconstruction of the urinary tract was before the metanephros undergoes hydronephrosis and when there was 0.016 cm³ urinary retention in the MNB. This timing suppressed the excessive dilatation of the kidney tubules of the transplanted metanephros, thereby reducing the progression of fibrosis. In fact, the main factors that limit the survival time are the remarkable tendency for hydronephrosis and the development of fibrosis in the transplanted MNB.

Currently, urinary tract reconstruction or extraction of the MNB or the metanephroi, which is performed 3 to 6 weeks after transplantation, depends on the animal species and the transplantation site [11–13, 19–21]. Metanephros weight gain stops at about 4 weeks after the transplantation [11, 13] and it has been reported that after development, the GFR of the metanephros or the creatinine clearance is about 0.3%–11% of normal kidneys, as the metanephros alone is a small organ [19, 21, 22]. For this reason, survival time does not differ in cases where one or several MNBs are transplanted [13].

In the present study, contrast-enhanced CT showed no enhancement of the transplanted metanephros or bladder; however, it allowed to evaluate the MNB volume. Due to the study methods involved in taking the measurements, we consider these results as accurate volume indices. A single intravenous dose of iohexol, the nonionic iodine-based contrast agent, undergoes rapid clearance from the blood of rats and translocates to the tissues [23]. It rapidly migrates to the kidneys and is distributed at high concentrations [23]. Yokote et al. [11] performed contrast-enhanced CT on MNBs to confirm urinary patency after SWPU, in which both kidneys were removed, the anastomotic ureter was ligated, and angiography was performed to visualize the recipient ureter [11]. In the present study, we took into consideration that native unilateral kidney was present in the recipients, resulting in the excretion of the contrast agent before it flowed into the MNB.

Similar to the contrast-enhanced CT examinations, ultrasonography detected the MNB in all the animals in the present study, and ultrasonography assessed the MNB morphology. Therefore, as the MNB was transplanted into the retroperitoneum, it was identified at an earlier stage from the time of transplantation due to the expansion of the retroperitoneal cavity. We also observed blood flow to the MNB using ultrasonography and observed the neovascularized vessels. Since urine production was previously confirmed by performing metanephros and MNB transplantation [11–13, 22], we hypothesized that the MNB bladder would be visualized with low echogenicity, making retrieval easier. In fetuses and infants with congenital hydronephrosis, ultrasonography can reveal the septum and cysts in the kidneys when severe hydronephrosis occurs [24, 25]. In the present study, it was also possible to evaluate the hydronephrotic metanephros without a urinary excretion pathway because the parenchyma was visualized as a vacuole with indistinct parenchyma, similar to severe hydronephrosis in a developing kidney. Spheroid volume measurements using ultrasonography have also been used in the kidney, thyroid, and prostate [26–30]. As the MNB volume determined using ultrasonography in this study strongly correlated with that determined using CT and the MNB volume was approximated despite being small, the volume calculation using the spheroid equation was considered accurate for the estimation of the MNB. Therefore, ultrasonography is recommended as a simple, minimally invasive, and useful method for MNB assessment compared to the contrast-enhanced CT examinations, which are invasive and involve radiation exposure.

In Phase 1, no difference was observed in fibrosis between MNB2 and MNB1, regardless of performing SWPU in MNB1. This suggests that by Day 28 after transplantation, many metanephroi had already experienced hydronephrosis, and the appropriate timing had passed despite urinary tract reconstruction. Indeed, previous reports refer to the onset of hydronephrosis by Day 21 after transplantation [11]. A comparison of the growth rates between MNB1 and MNB2 showed no significant difference in the volume; however, the detection of MNB2 using ultrasonography was higher during Day 7. A plurality of blood vessels around the MNB1 and MNB2 was confirmed using a color Doppler method on Day 14 after transplantation. It has been confirmed that the transplanted metanephros regenerates recipient-derived blood vessels and is chimerized with the vessels [31]. Angiogenesis involves vascular endothelial growth factor (VEGF), platelet-derived growth factor (PDGF), and fibroblast growth factor

(FGF) [32]. These angiogenic factors have important roles in tissue ischemia and angiogenesis. One report suggests that the metanephros receives spatial direction for capillary development from VEGF [33] and could be involved in the vasculature around the MNB. However, it must be considered that VEGF is also involved in fibrosis, which promotes the growth of grafts and may further exacerbate fibrosis during hydronephrosis [34, 35]. In a unilateral ureteric obstruction (UUO) model of progressive kidney injury, an angiogenic response was observed early and the endothelial cells proliferated. However, this led to endothelial cell loss after the 4$^{th}$ day, the neovascularized capillaries disappeared, and the tissue fell into an ischemic state again [36]. The MNB2 could be affected by MNB1 angiogenesis because it was implanted just above the MNB1 in the retroperitoneal cavity. Furthermore, the higher detection rate of the MNB2 than that of the MNB1 on Day 7 after each MNB was transplanted and the presence of tubular dilation on Day 28 after MNB2 was transplanted suggest that the neovascularization of the MNB1 may also affect and promote the growth of the MNB2. One of the other reasons for the higher detection rate of MNB2 on Day 7 may be that one of the recipient's kidneys was removed during the MNB2 transplantation. Studies in fetal ewes have shown that in the event of a sudden loss of unilateral kidney function due to an obstruction in the unilateral ureter, the remaining kidney plays a compensatory role, resulting in enlargement of the remaining kidney even in the fetal period [37]. It has also been reported that when the mother's kidney is removed during gestation in the rat, the fetal glomerular volume increases, and the development of glomeruli is greater than that in an unextracted sham-operated fetal kidney [38]. These results suggest that a growth period of 4 weeks after transplantation may be sufficient to develop the hydronephrotic metanephros of the MNB2, the degree of MNB growth is not constant when multiple grafts are transplanted, and that growth may vary greatly between the MNBs.

In Phase 2, only one MNB was transplanted, and the MNB growth rate and appearance of the MNB bladder were observed in detail over time to establish an appropriate time index for performing SWPU. Even when only one MNB was transplanted, the growth rate varied among individuals, as interpreted from the results of Phase 1. The particularly well-developed MNBs (38.9%) showed bladder dilation within 21 days after transplantation. By Day 21, more than half of the MNBs reached the criteria for SWPU, suggesting that SWPU should be done earlier than previously reported. Furthermore, there was a poorly developed MNB with no change in appearance or change in the size of the bladder from Day 21. Histopathological examination did not show excessive tubular dilation or progression of fibrosis as observed in Phase 1 in the MNB resected within the appropriate timeframe for SWPU. Obstruction usually causes increased ureteral/kidney pelvis pressure and follows tubular dilatation in the kidney. This increase in pressure stimulates tubular epithelial cells and causes fibrosis to progress due to epithelial-mesenchymal transition [39]. In a previous study using the rat UUO model, the expression of TGF-β1 was detected immediately after the obstruction of the urinary tract and the expression of vimentin and myofibroblasts 2 days after the obstruction [40]. Similarly, in an experiment conducted on the organogenesis stage using the rat neonatal UUO model, where the growth rate decreased by 30%, and blood pressure, GFR, urine flow, and sodium/potassium excretion decreased even after unblocking 5 days after the obstruction [15]. Therefore, it has been shown that the UUO model is particularly susceptible to long-term effects from events occurring immediately after kidney formation [14, 15].

Although the transplanted MNB cannot be completely explained using the neonatal UUO model, if we consider the period after transplantation as being akin to the neonatal period, the period of occlusion can be expected to be similarly involved in the growth of the transplanted metanephros. In this study, observations with ultrasonography were performed every other day and some rats showed rapid bladder dilation. Rats have only 10% of the nephrons formed

at birth, and it is said that kidney formation is completed within the first week after birth [15]. The rats used in this study were taken on an embryonic Day 15, and the birth of a rat is usually after 21 to 23 days gestation on average; therefore, the morphological formation of the kidney completed 2 weeks after transplantation and the rapid development of urine in the MNB 2 to 3 weeks after transplantation would be considered as normal development. Therefore, we believe that MNBs that grow well may grow at the same growth rate as the *in vivo* organs of a normal fetal rat. Hence, it is important to observe and evaluate the transplanted bladder dilation daily starting from the second week when the MNB grows rapidly in the rat. However, these results may change in other animal species and further investigation is needed.

The GFR measurement of the transplanted metanephros showed a low value of 3% to 11% of the normal capacity in the present study. Although the same measurement method was not used, we predicted that the SP group would be within 5% of normal. This estimate was based on a previous report where the metanephros grew about 90 to 116 days after reconstructing the urinary tract [22] and it was thought that a certain amount of tissue was present. Urinary tract reconstruction for the metanephros was performed at a time determined with the naked eye and reconstruction was performed through an end-to-end anastomosis of the ureter. Therefore, the metanephros may have been exposed to long-term obstruction. It has been shown that prolonged obstruction results in persistent dilation of the tubule and glomerular Bowman's capsule and results in the production of inflammatory cytokines such as TGF-β from the tubular cells [41]. TGF-β1 is expressed in tissues such as the hematopoietic tissue, endothelial tissue, and bone tissue in developing embryos and acts as an important growth factor in heart formation [42, 43]. However, TGF-β1 is also deeply involved in fibrosis, causing epithelial to mesenchymal transition in tubular cells in the kidney, inducing expression of vimentin, collagen, and alpha-smooth muscle actin [41]. In addition, TGF-β1 targets protease inhibitor-1 (PAI-1) in the proximal tubular epithelial cells and stromal fibroblasts, and the transcription factor p53 replicates through this mechanism. It causes an aging state and induces cell growth inhibition and apoptosis [41, 44–46]. The TGF-β1 expression has also been reported in fetal metanephros. In the present study, the SWPU inhibited TGF-β1 expression due to the persistent inhibition of tubular dilatation. It is presumed that the suppressed tissue was replaced by collagen and vimentin. Therefore, the fact that GFR could be measured in the tissues approximately 60 days after transplantation suggests additional development of GFR value in long-term observation, and this adds to our knowledge regarding organ growth.

Despite these findings, there are some limitations to this study to consider. First, we only performed diagnostic imaging-based assessments and did not assess other physiological aspects such as renin and erythropoietin activity. Second, because we assessed the MNB for a short period (60 days after transplantation), we did not perform a long-term assessment of the function and morphology of the transplants. These issues need to be elucidated further in future studies.

## Conclusions

In conclusion, in this study, we successfully observed the time of MNB growth in detail using an ultrasonographic examination. As a result, we believe that the appropriate timing for urinary tract reconstruction in rats using SWPU is when the metanephros has not undergone hydronephrosis and there is urinary retention of 0.016 cm$^3$ in the MNB. This method suppresses the excessive dilatation of the kidney tubules of the transplanted metanephros, seemingly reducing the progression of fibrosis. We believe this evidence will greatly contribute to the evaluation of MNB development and urinary tract reconstruction in xenotransplantation and human clinical practice. In addition, the results may be applicable to other transplanted

organs with further study, thus emphasizing the importance of evaluating the morphology and function of small grafts with minimally invasive methods when considering transplantation into human patients in the future.

## Supporting information

**S1 Fig. Stepwise peristaltic ureter (SWPU) system.** The metanephros with bladder (MNB) is transplanted in the recipient and allowed to grow. The ureter of the recipient is anastomosed to the MNB bladder where urine has accumulated. In this manner, urinary excretion from the MNB can be measured.
(PDF)

**S1 Table. Glomerular filtration rate (GFR) measurements in healthy adult rats using inulin clearance.** Measurements are performed under the same conditions as those maintained for the GFR measurement method in the metanephros with the bladder.
(PDF)

## Acknowledgments

We would like to thank Dr. Satoshi Kameshima for advice on the experiment and Editage (www.editage.com) for English language editing.

## Author Contributions

**Conceptualization:** Kotaro Nishi, Takashi Yokoo, Satomi Iwai.

**Data curation:** Kotaro Nishi.

**Formal analysis:** Kotaro Nishi, Takafumi Haji, Takuya Matsumoto, Chisato Hayakawa.

**Funding acquisition:** Kenichi Maeda, Shozo Okano, Takashi Yokoo, Satomi Iwai.

**Investigation:** Kotaro Nishi, Takafumi Haji, Takuya Matsumoto, Chisato Hayakawa.

**Methodology:** Kotaro Nishi, Takashi Yokoo, Satomi Iwai.

**Project administration:** Kotaro Nishi, Takashi Yokoo, Satomi Iwai.

**Software:** Kenichi Maeda.

**Supervision:** Kotaro Nishi, Shozo Okano, Satomi Iwai.

**Validation:** Satomi Iwai.

**Writing – original draft:** Kotaro Nishi.

**Writing – review & editing:** Satomi Iwai.

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
