## [Decision Letter · Decision Letter 0]

30 Jun 2020

PONE-D-20-07683

Reconstruction of the urinary tract at the appropriate time reduces fibrosis of the metanephros in rats as judged by imaging

PLOS ONE

Dear Dr. Iwai,

Thank you for submitting your manuscript to PLOS ONE. After careful consideration, we feel that it has merit but does not fully meet PLOS ONE’s publication criteria as it currently stands. Therefore, we invite you to submit a revised version of the manuscript that addresses the points raised during the review process.

Please submit your revised manuscript by the 08/15/ 2020. If you will need more time than this to complete your revisions, please reply to this message or contact the journal office at plosone@plos.org. Please include the following items when submitting your revised manuscript:

We look forward to receiving your revised manuscript.

Kind regards,

Franziska Theilig, Prof.

Academic Editor

PLOS ONE

Journal Requirements:

Additional Editor Comments (if provided):

Reviewers' comments:

Reviewer's Responses to Questions

**Comments to the Author**

1. Is the manuscript technically sound, and do the data support the conclusions?

Reviewer #1: Partly

Reviewer #2: Yes

Reviewer #3: Yes

2. Has the statistical analysis been performed appropriately and rigorously? 

Reviewer #1: No

Reviewer #2: Yes

Reviewer #3: Yes

3. Have the authors made all data underlying the findings in their manuscript fully available?

Reviewer #1: Yes

Reviewer #2: Yes

Reviewer #3: Yes

4. Is the manuscript presented in an intelligible fashion and written in standard English?

Reviewer #1: No

Reviewer #2: Yes

Reviewer #3: No

5. Review Comments to the Author

Reviewer #1: Nishi et al. monitored transplanted rat embryonic kidneys by CT and ultrasonography, and found that ureter reconstruction should be performed within 3 weeks after transplantation. When the optimal timing was determined by ultrasonography, the resultant kidneys showed less fibrosis.

The data presentation is too preliminary. There are as many as 10 Figures, but the number of Figures can be dramatically reduced. The text should be extensively re-organized to convey the experimental designs and results in a more understandable manner. The Abstract and Introduction section sound as though the authors pioneered iPS cell-derived organoids, and many inappropriate references have been cited. In addition, the text contains so many grammatical errors that it is difficult to understand the logical flow of the Results and Discussion sections.

Major comments

1. The whole manuscript should be extensively proofread by a native English speaker. In the present form, it is difficult to evaluate the correctness of the results or the logical flow of the discussions. Thus, I have not commented on the Discussion section in this round of review.

2. The experimental designs (Experiments 1, 2, and 3) and sample definitions (such as MNB 2, SP, and 28UR) should be included in the Results section, so that readers can understand the aims of the experiments. The descriptions in the Methods section should be limited to detailed technical issues. Having said that, the present Methods section lacks some important points, such as how the areas of tubule dilatation or fibrosis were calculated per slide and how many slides were analyzed per kidney.

3. The number of Figures should be reduced. Furthermore, the experimental designs, which are currently presented as Supplementary Figures, can be incorporated into the main Figures. For example, the design for Experiment 1, Fig. 1, and Fig. 3 can be combined into a single Figure. Likewise, the design for Experiment 2, Fig. 4, and Fig. 5 can be combined, as can the design for Experiment 3, Fig. 6, and Fig. 7. Figs. 8–10 can also be combined.　

4. The TUNEL staining in Fig. 10 is too weak and hardly different from the background signals. Replace with more convincing photos or delete the whole figure.

5. It is not clear how many samples were processed in each experiment. Clearly indicate the sample numbers in each Figure and Table or their corresponding legends.

6. Explain each figure panel one by one in the text. In the present form, the contribution of each panel to the logical flow of the text remains unclear.

7. In the Abstract, the authors claimed that they generated iPS cell-derived kidneys, but this is not supported by the literature. In the Introduction section, the authors claim that they generated iPS cell-derived kidney tissues, but the cited references only report data using mouse, rat, or mesenchymal stem cells. If this is done intentionally, it is a serious violation against scientific rules and ethics. The authors should avoid presenting a misleading Introduction section by correctly explaining their own achievements and fairly citing the key papers on iPS cell-derived organoids. At present, only two references are cited (references #3 and #4), and they are misinterpreted.

8. The title does not convey the precise message. This manuscript deals with transplanted kidneys. Consider revising the title.

Minor comments

1. Fig. 2 is actually a Table. Therefore, present it as a Table. Alternatively, it can be incorporated into other Figures.

2. Do Figs. 7A and 8A represent the data before 3 weeks or at 8 weeks after transplantation? The Figure Legends suggest the former, but the actual Figures appear to indicate the latter.

3. Table 1 is not cited in the text.

4. Figs. S1 and S2 are mislabeled.

5. The Tajiri Scientific Reports paper remains uncited on page 5.

6. The explanation of Fig. 5 in the text appears to be opposite to the actual data.

7. An incorrect paper (reference #2) is cited on page 26.

Reviewer #2: The nephron progenitor (MNB: Metanephros with bladder) developed and started producing urine following transplantation in vivo. However, metanephros can cause hydronephrosis because transplantation alone does not provide a route for excretion of the produced urine. This study examined the appropriate time index for urinary tract reconstruction (stepwise peristaltic ureter system: SWPU). Clinical and general imaging methods, particularly, ultrasonography have been used because they help to easily determine the condition of the body and are less invasive. The authors conclude that the appropriate time for SWPU was revealed by ultrasonography and the progression of fibrosis of the transplanted metanephros could be reduced. The accuracy of the experiment is high and the results are credible. I would like to offer some minor comments.

1). Is it able to apply for clinical application although human body size is larger than that of rats?

2). What do you think about the function other than the production of urine, for instance, renin?

Reviewer #3: "Reconstruction of the urinary tract at the appropriate time reduces fibrosis of the metanephros in rats as judged by imaging" (PONE-D-20-07683)

In this manuscript, Nishi K. et al. evaluated the growth of transplanted metanephros with bladder (MNB) by comparing computed tomography and ultrasonography and demonstrated that ultrasonographic findings showed high correlations with computed tomographic findings. The authors found that, although the MNB growth after the transplantation differed among individuals, most MNBs reach the appropriate period for urinary tract reconstruction within 3 weeks after transplantation. The authors concluded that optimizing the stepwise peristaltic ureter system anastomosis by ultrasonography reduces long-term tubular dilation of the metanephros, thereby decreasing fibrosis caused by transforming growth factor-β1. Overall, the data seem to support their conclusions. This study would significantly contribute to the regeneration of kidneys by the authors’ approach.

Major points:

1) There seems to be many grammatical and typographical errors and inaccurate descriptions, which makes this manuscript difficult to understand. The English should be revised throughout the manuscript by a native English speaker.

2) One of the two criterion for the appropriate timing of urinary tract reconstruction used in this study is the MNB bladder volume larger than 0.016 cm3. This reviewer is wondering how they decided the specific volume (0.016 cm3). The data comparing 0.016 cm3 and other volumes or the explanation of the process to make the criterion should be added.

3) The images of Figure 10A are unclear. It is now difficult to find the TUNEL (+) apoptotic cells and glomeruli. The clearer images are required.

Minor points:

1) Abstract, line 24. “iPS-derived generated kidney” should be replaced by “iPS cell-derived regenerated kidney”.

2) Abstract, line 37. “transforming growth factor-β” should be replaced by “transforming growth factor-β1.”

3) Abstract, line 40. “kidney generation” should be replaced by “kidney regeneration.”

4) Introduction, line 50. “a need treatment for alternatives” is inaccurate. Should the description be “a need for alternative treatment”?

5) Introduction, line 52. “Takasato et al., [3,4]” is wrong. The ref. [4] was published by another group.

6) Introduction, line 71. “this success” should be replaced by “these successes”.

7) Introduction, line 76. SWPU is shown now in “S2 Fig” but not in “S1 Fig”.

8) Materials and methods, line 234. “20 taken” should be replaced by “20 images taken”.

9) Results, lines 303 and 304. Is this data “data not shown”?

10) Results, lines 314-320. Are these data “data not shown”?

11) Results, lines 341-343. Is this sentence accurate? Figure 4A seems to indicate that the MNBs removed 21 or more days after transplantation had significantly more severe tubular dilation than those removed less than 21 days after transplantation.

12) Results, line 378. “tubular” should be replaced by “tubules”.

13) Results, lines 396 and 398. “interstitial” should be replaced by “interstitium”.

14) Figure 10 legends do not correspond to the figures. The legend B) explains lower panels in Figure A, and the legend describing Figure B is missing.

6. PLOS authors have the option to publish the peer review history of their article (what does this mean?). If published, this will include your full peer review and any attached files.

Reviewer #1: No

Reviewer #2: **Yes: **Motoaki Sano

Reviewer #3: No

---

## [Author Response · Author response to Decision Letter 0]

2 Oct 2020

Dear All Reviewers,

We are grateful for the expert comments on this issue and the request for a revised version.

We have copy and pasted all reviewer’s comments below, and made every attempt to incorporate these suggestions as thoroughly as possible.

I hope that revised manuscript is now acceptable for publication.

Best regards,

Satomi Iwai

Associate professor

Laboratory of Small Animal Surgery 2

School of Veterinary Medicine

Kitasato University

35-1, Higashi23, Towada-city, Aomori, 034-8628

E-mail: iwai@vmas.kitasato-u.ac.jp

---

## [Decision Letter · Decision Letter 1]

21 Oct 2020

PONE-D-20-07683R1

Timing urinary tract reconstruction in rats to avoid hydronephrosis and fibrosis in the transplanted fetal metanephros as assessed by imaging

PLOS ONE

Dear Dr. Iwai,

Thank you for submitting your manuscript to PLOS ONE. After careful consideration, we feel that it has merit but does not fully meet PLOS ONE’s publication criteria as it currently stands. Therefore, we invite you to submit a revised version of the manuscript that addresses the points raised during the review process.

We look forward to receiving your revised manuscript.

Kind regards,

Franziska Theilig, Prof.

Academic Editor

PLOS ONE

Reviewers' comments:

Reviewer's Responses to Questions

**Comments to the Author**

1. If the authors have adequately addressed your comments raised in a previous round of review and you feel that this manuscript is now acceptable for publication, you may indicate that here to bypass the “Comments to the Author” section, enter your conflict of interest statement in the “Confidential to Editor” section, and submit your "Accept" recommendation.

Reviewer #1: (No Response)

Reviewer #2: All comments have been addressed

Reviewer #3: All comments have been addressed

2. Is the manuscript technically sound, and do the data support the conclusions?

Reviewer #1: Yes

Reviewer #2: Yes

Reviewer #3: Yes

3. Has the statistical analysis been performed appropriately and rigorously? 

Reviewer #1: Yes

Reviewer #2: Yes

Reviewer #3: Yes

4. Have the authors made all data underlying the findings in their manuscript fully available?

Reviewer #1: Yes

Reviewer #2: Yes

Reviewer #3: Yes

5. Is the manuscript presented in an intelligible fashion and written in standard English?

Reviewer #1: Yes

Reviewer #2: Yes

Reviewer #3: No

6. Review Comments to the Author

Reviewer #1: The authors have substantially improved the manuscript. However, there are still some points that remain to be addressed.

1. Abstract (page 4, line 45): Add “possibly” to the sentence, because the authors did not unequivocally prove that renal fibrosis was caused by TGF-b1.

2. Describe the references precisely. References 5–7 on page 5 used mesenchymal stem cells, and not induced stem cells such as iPS cells. Reference 10 simply shows that NPCs can be induced from patient-derived iPS cells and does not demonstrate the formation of interspecies chimeric nephrons.

3. The authors retracted the figure showing apoptosis, because they were unable to provide a more convincing stained image than that in the first submission, in which the positive staining was hardly distinguishable from the background staining. Nonetheless, the quantitative data based on such unreliable staining are still present in the manuscript and figure. If the authors consider the quantitative data to be reliable, that is fine. Otherwise, Figure 5E and 5F, as well as the related descriptions in the text, should be deleted.

4. Figure 5A and 5C should be presented in a larger size, so that readers can recognize the abnormal histology in detail.

5. Although the initial sample numbers are described for the schematic in Figure 1, the relevant sample numbers should also be described for all figures and tables that contain statistical analyses. For example, Table 3, Figure 3B and 3C, and Figure 5B and 5D.

Reviewer #2: As for the question I posed, it has been adequately answered. There are no further concerns and requirements.

Reviewer #3: "Reconstruction of the urinary tract at the appropriate time reduces fibrosis of the metanephros in rats as judged by imaging" (PONE-D-20-07683R1)

The authors have addressed my concerns. I have following minor points to be addressed.

Minor points:

1) Introduction, line 59. “alternatives treatment” should be “alternative treatment”.

2) Introduction, lines 69 to 70. “induced stem cells” should be “induced pluripotent stem cells”.

3) Introduction, line 74. “NPCs with induced pluripotent stem cells” should be “NPCs differentiated from induced pluripotent stem cells”.

4) Materials and method, lines 144 to 147. While it is described that the first group had left recipient kidney removed, Fig. 1A indicates that right recipient kidney is removed and used for SWPU.

5) Materials and method, lines 150 to 152. While it is described that the MNB2 is located to the head side of MNB1, Fig. 1A indicates that the opposite positions of MNB1 and MNB2.

6) Materials and method, lines 235 to 238. The two sentences seem grammatically incorrect and should be revised.

7) Materials and methods, line 238. “Day28” should be replaced by “Day56”.

8) Results, lines 382 to 394. Fig. 3A should be cited in the text.

9) Results, line 396. “Histopathological” should be “histopathological”.

10) Results, line 455. “Above figure” should be “Upper figure”.

7. PLOS authors have the option to publish the peer review history of their article (what does this mean?). If published, this will include your full peer review and any attached files.

Reviewer #1: No

Reviewer #2: No

Reviewer #3: No

---

## [Author Response · Author response to Decision Letter 1]

6 Nov 2020

November 6, 2020

Dr. Franziska Theilig

Academic Editor

PLOS ONE

Dear Editor:

Thank you again for inviting us to submit a revised draft of our manuscript entitled, “Timing urinary tract reconstruction in rats to avoid hydronephrosis and fibrosis in the transplanted fetal metanephros as assessed using imaging.” The original submission to PLOS ONE was on Mar 17th, 2020. The manuscript ID number is PONE-D-20-07683.

We appreciate the time and effort you and each of the reviewers have dedicated to provide insightful feedback on ways to strengthen our paper. Thus, it is with great pleasure that we resubmit our article for further consideration with the incorporated changes that reflect the detailed suggestions you have graciously provided. We hope that our edits, highlighted in the manuscript in red font, and the response to reviewers provided below have satisfactorily addressed all of the issues noted.

Thank you for your consideration. I look forward to hearing from you.

Sincerely,

Satomi Iwai

Associate professor

Laboratory of Small Animal Surgery 2

School of Veterinary Medicine

Kitasato University

35-1, Higashi23, Towada-city, Aomori, 034-8628

E-mail: iwai@vmas.kitasato-u.ac.jp

Response to Reviewer 1: I am very grateful for the valuable time and effort you have put into reviewing my paper. I have carefully considered your comments and made the relevant changes in the manuscript.

1. Abstract (page 4, line 45): Add “possibly” to the sentence, because the authors did not unequivocally prove that renal fibrosis was caused by TGF-b1.

Response: Thank you for your suggestion. I have added the word. (Line 39)

2. Describe the references precisely. References 5–7 on page 5 used mesenchymal stem cells, and not induced stem cells such as iPS cells. Reference 10 simply shows that NPCs can be induced from patient-derived iPS cells and does not demonstrate the formation of interspecies chimeric nephrons.

Response: Thank you for your comment. I have corrected the text. Since the content was misleading to the reader, a description of iPS cells has been added. (Lines 66-76)

3. The authors retracted the figure showing apoptosis, because they were unable to provide a more convincing stained image than that in the first submission, in which the positive staining was hardly distinguishable from the background staining. Nonetheless, the quantitative data based on such unreliable staining are still present in the manuscript and figure. If the authors consider the quantitative data to be reliable, that is fine. Otherwise, Figure 5E and 5F, as well as the related descriptions in the text, should be deleted.

Response: We agree with your assessment. We believe that the data are reliable because we had the images checked daily by individuals who are experts in checking details of stained images. However, the image was certainly unclear, so I have enlarged the image to make it easier to understand. I would appreciate it if you could check it. (Lines 437-442, Figure 5E)

4. Figure 5A and 5C should be presented in a larger size, so that readers can recognize the abnormal histology in detail.

Response: Thank you for your suggestion. Figures 5 A and C were corrected by enlarging the images. 

5. Although the initial sample numbers are described for the schematic in Figure 1, the relevant sample numbers should also be described for all figures and tables that contain statistical analyses. For example, Table 3, Figure 3B and 3C, and Figure 5B and 5D.

Response: As you pointed out, the number of samples was included in the figures and tables. (Table 3, Figures 3B and C and 5B and D)

Response to Reviewer 2: Thank you very much for spending your valuable time to review my manuscript. 

Response to Reviewer 3: Thank you for taking the time to comment on our manuscript again. I have made corrections in consideration of your comments. 

Minor points:

1) Introduction, line 59. “alternatives treatment” should be “alternative treatment”.

Response: Thank you for your keen observation. This has been revised. (Line 54)

2) Introduction, lines 69 to 70. “induced stem cells” should be “induced pluripotent stem cells”.

Response: Thank you for your comment. This has been changed because it refers to mesenchymal stem cells. (Lines 66-67)

3) Introduction, line 74. “NPCs with induced pluripotent stem cells” should be “NPCs differentiated from induced pluripotent stem cells”.

Response: We agree with your assessment. This has been revised in Line 72-73.

4) Materials and method, lines 144 to 147. While it is described that the first group had left recipient kidney removed, Fig. 1A indicates that right recipient kidney is removed and used for SWPU.

Response: Thank you for providing these insights. The entire Figure 1 was confirmed again and the statement about SWPU has been revised. 

5) Materials and method, lines 150 to 152. While it is described that the MNB2 is located to the head side of MNB1, Fig. 1A indicates that the opposite positions of MNB1 and MNB2.

Response: We agree with you and have incorporated this suggestion throughout the Figure.

6) Materials and method, lines 235 to 238. The two sentences seem grammatically incorrect and should be revised.

Response: I have rechecked the sentences; the entire manuscript has been edited by a native English speaker (Lines 238 - 242)

7) Materials and methods, line 238. “Day28” should be replaced by “Day56”.

Response: Thank you for your careful reading. I have revised it. (Line 241)

8) Results, lines 382 to 394. Fig. 3A should be cited in the text.

Response: Thank you for your comment. Fig 3A has been cited. (Line 392)

9) Results, line 396. “Histopathological” should be “histopathological”.

Response: I have made the revision. (Line 401)

10) Results, line 455. “Above figure” should be “Upper figure”.

Response: Thank you for your suggestion. I have made the revision. (Line 461)

---

## [Decision Letter · Decision Letter 2]

1 Dec 2020

PONE-D-20-07683R2

Timing urinary tract reconstruction in rats to avoid hydronephrosis and fibrosis in the transplanted fetal metanephros as assessed using imaging

PLOS ONE

Dear Dr. Iwai,

Thank you for submitting your manuscript to PLOS ONE. After careful consideration, we feel that it has merit but does not fully meet PLOS ONE’s publication criteria as it currently stands. Therefore, we invite you to submit a revised version of the manuscript that addresses the points raised during the review process.

Unfortunately, there are still some mistakes in the Figure 5 E. Please change the phrase and the Figure 5 E.

We look forward to receiving your revised manuscript.

Kind regards,

Franziska Theilig, Prof.

Academic Editor

PLOS ONE

Reviewers' comments:

Reviewer's Responses to Questions

**Comments to the Author**

1. If the authors have adequately addressed your comments raised in a previous round of review and you feel that this manuscript is now acceptable for publication, you may indicate that here to bypass the “Comments to the Author” section, enter your conflict of interest statement in the “Confidential to Editor” section, and submit your "Accept" recommendation.

Reviewer #1: All comments have been addressed

Reviewer #2: All comments have been addressed

Reviewer #3: All comments have been addressed

2. Is the manuscript technically sound, and do the data support the conclusions?

Reviewer #1: Yes

Reviewer #2: Yes

Reviewer #3: Yes

3. Has the statistical analysis been performed appropriately and rigorously? 

Reviewer #1: Yes

Reviewer #2: Yes

Reviewer #3: Yes

4. Have the authors made all data underlying the findings in their manuscript fully available?

Reviewer #1: Yes

Reviewer #2: Yes

Reviewer #3: Yes

5. Is the manuscript presented in an intelligible fashion and written in standard English?

Reviewer #1: Yes

Reviewer #2: Yes

Reviewer #3: No

6. Review Comments to the Author

Reviewer #1: The authors have adequately addressed my concerns.

Minor corrections

1. Abstract (Line 39 on page 4): thereby decreasing the fibrosis caused possibly by transforming growth factor-b1.

2. Line 442 on page 29: Fig 5G (not Figs 5G).

Reviewer #2: I have no further concerns about this manuscript entitled Timing urinary tract reconstruction in rats to avoid hydronephrosis and fibrosis in the transplanted fetal metanephros as assessed using imaging.

Congratulations on a great job.

Reviewer #3: The authors have addressed my concerns. I have following minor points to be addressed.

Minor points:

1) Discussion, lines 528-529. “the transplanted metanephros regenerates recipient-derived blood vessels and is chimerized with the donor’s metanephros”. The authors should confirm whether this sentence is grammatically correct. I think “the transplanted metanephros regenerates recipient-derived blood vessels and is chimerized with the vessels” is correct.

2) Figure 5E. TUNEL(+) cells are more frequently observed in SP group than in 28 UR group, which is opposite to the results of Figure 5F. The authors should replace Figure 5E with more representative images.

7. PLOS authors have the option to publish the peer review history of their article (what does this mean?). If published, this will include your full peer review and any attached files.

Reviewer #1: No

Reviewer #2: No

Reviewer #3: No

---

## [Author Response · Author response to Decision Letter 2]

15 Dec 2020

December 15, 2020

Dr. Franziska Theilig

Academic Editor

PLOS ONE

Dear Editor:

Thank you again for inviting us to submit a revised draft of our manuscript titled, “Timing urinary tract reconstruction in rats to avoid hydronephrosis and fibrosis in the transplanted fetal metanephros as assessed using imaging.” The initial draft was submitted to PLOS ONE on March 17, 2020, with manuscript ID number PONE-D-20-07683.

We appreciate the time and effort you and the reviewers have dedicated in providing insightful feedback on ways to strengthen our paper. We believe that our edits, highlighted in the manuscript in red, and the responses to reviewer comments provided below have satisfactorily addressed all the issues noted.

Thank you for your consideration. I look forward to hearing from you.

Sincerely,

Satomi Iwai

Associate professor

Laboratory of Small Animal Surgery 2

School of Veterinary Medicine

Kitasato University

35-1, Higashi23, Towada-city, Aomori, 034-8628

E-mail: iwai@vmas.kitasato-u.ac.jp

 

Response to Reviewer 1: Thank you for your valuable comments. We learned a lot from your comment. Please check our revised version.

Minor corrections:

1. Abstract (Line 39 on page 4): thereby decreasing the fibrosis caused possibly by transforming growth factor-b1.

Response: Thank you for your suggestion. We have revised the sentence. (Line 40)

2. Line 442 on page 29: Fig 5G (not Figs 5G).

Response: Thank you for your careful review. We have corrected the word. (Lines 444-445)

Response to Reviewer 2: Thank you for taking time to review our manuscript. 

Response to Reviewer 3: Thank you for taking the time to review our manuscript again. We have made corrections per your comments. For English, we asked for native proofreading again. Thank you for your confirmation.

Minor points:

1) Discussion, lines 528-529. “the transplanted metanephros regenerates recipient-derived blood vessels and is chimerized with the donor’s metanephros”. The authors should confirm whether this sentence is grammatically correct. I think “the transplanted metanephros regenerates recipient-derived blood vessels and is chimerized with the vessels” is correct.

Response: Thank you for your comment. This has been revised as per your comment. (Line 530–532)

2) Figure 5E. TUNEL(+) cells are more frequently observed in SP group than in 28 UR group, which is opposite to the results of Figure 5F. The authors should replace Figure 5E with more representative images.

Response: Thank you for your comment. In Fig 5E, the results of the SP and 28UR groups were reversed. We have corrected this. We apologize for this oversight on our part.

---

## [Decision Letter · Decision Letter 3]

30 Dec 2020

Timing urinary tract reconstruction in rats to avoid hydronephrosis and fibrosis in the transplanted fetal metanephros as assessed using imaging

PONE-D-20-07683R3

Dear Dr. Iwai,

We’re pleased to inform you that your manuscript has been judged scientifically suitable for publication and will be formally accepted for publication once it meets all outstanding technical requirements.

Kind regards,

Franziska Theilig, Prof.

Academic Editor

PLOS ONE

Reviewers' comments:

Reviewer's Responses to Questions

**Comments to the Author**

1. If the authors have adequately addressed your comments raised in a previous round of review and you feel that this manuscript is now acceptable for publication, you may indicate that here to bypass the “Comments to the Author” section, enter your conflict of interest statement in the “Confidential to Editor” section, and submit your "Accept" recommendation.

Reviewer #3: (No Response)

2. Is the manuscript technically sound, and do the data support the conclusions?

Reviewer #3: (No Response)

3. Has the statistical analysis been performed appropriately and rigorously? 

Reviewer #3: (No Response)

4. Have the authors made all data underlying the findings in their manuscript fully available?

Reviewer #3: (No Response)

5. Is the manuscript presented in an intelligible fashion and written in standard English?

Reviewer #3: (No Response)

6. Review Comments to the Author

Reviewer #3: The authors have addressed my concerns. I have no further concerns except for the following point.

It is described that red and yellow arrows indicate MNB1 and MNB2, respectively, in lines 319 – 320 and that MNB2 was transplanted to the head side of MNB1 in lines 154 -155. However, it seems that red MNB1 is located to the head side of yellow MNB2 in Figure 2A. The authors should confirm the spatial relationship and the description.

7. PLOS authors have the option to publish the peer review history of their article (what does this mean?). If published, this will include your full peer review and any attached files.

Reviewer #3: No

---

## [Editor Report · Acceptance letter]

6 Jan 2021

PONE-D-20-07683R3 

Timing urinary tract reconstruction in rats to avoid hydronephrosis and fibrosis in the transplanted fetal metanephros as assessed using imaging 

Dear Dr. Iwai:

I'm pleased to inform you that your manuscript has been deemed suitable for publication in PLOS ONE. Congratulations! Your manuscript is now with our production department. 

Kind regards, 

on behalf of

Dr. Franziska Theilig 

Academic Editor

PLOS ONE